# Corrosion of Stirred Electrochemical Nano-Crystalline Hydroxyapatite (HA) Coatings on Ti6Al4V

**DOI:** 10.3390/ma15238609

**Published:** 2022-12-02

**Authors:** Narayanan Ramaswamy, Venkatachalam Gopalan, Tae Yub Kwon

**Affiliations:** 1School of Mechanical Engineering, VIT Chennai, Chennai 600 127, India; 2Centre for Innovation and Product Development, VIT Chennai, Chennai 600 127, India; 3Department of Dental Biomaterials, School of Dentistry, Kyungpook National University, Daegu 41566, Republic of Korea

**Keywords:** Ti6Al4V, ultrasonics, coating, nano-crystalline, hydroxyapatite, corrosion

## Abstract

Ti6Al4V substrates were electrochemically deposited with nano-crystalline hydroxyapatite (HA) from aqueous electrolytes. Cathodic HA coatings were obtained when the electrolyte was stirred using ultrasonic vibration. Two current densities of 20 mA/cm^2^ and 50 mA/cm^2^ were employed. Polarization and electrochemical impedance spectroscopy (EIS) were the techniques used to estimate the corrosion of coatings in simulated body fluid (SBF). The results indicate good corrosion resistance for the coating obtained at 50 mA/cm^2^ from ultrasonic stirring of the electrolyte.

## 1. Introduction

Titanium systems are widely used for body implant applications (failed hard tissue) due to their (i) high corrosion resistance and (ii) good bone compatibility [1,2,3,4]. This excellent level of biocompatibility is due to the presence of stable and protective oxide formed on titanium naturally [4]. Among the members of the titanium family, Ti6Al4V has received much attention for use in bone implants, primarily because of its biocompatibility and fatigue properties. Use of this alloy is not without disadvantages, as it may release the constituent metallic ions inside the body. The release of even a small quantity of these ions causes irritation in the localized region of the implant [2]. Additionally, these cell and tissue responses are influenced by the implant surface topography or roughness [1].

Calcium phosphate is a widely used ceramic system in clinical (orthopedic/dental) applications. This is predominantly due to the excellent biocompatibility exhibited by the calcium phosphates inside the body [5]. One of the types of calcium phosphate, hydroxyapatite (HA), is highly biocompatible and non-toxic, has non-inflammatory qualities, and increases the rate of bone formation in the pores. Biological HA is different from the mineral type as it consists of many variations, such as replacements, vacancies, non-stoichiometry content, etc. First Principle Modeling and Local Density Approximation methods [6] indicate a tolerable level of HA resistance to heating, microwave radiation, hydrogenation, and synchrotron radiation.

However, fracture toughness of HA is low compared with bone [7], indicating poor performance in tensile loading conditions. When X-ray radiation is incident on the mineral/organic matrix interaction of bone tissue, changes on a molecular level are observed and are correlated with alterations in the mechanical properties of the bone samples [8]. The outcomes of HA may also depend on its resistance to aging, including radiation. Reinforcements may be added to improve the properties of bone mineral. For example, ZrO_2_ and MgO nanoparticle additions made to bone cement blends (poly methyl methacrylate, also known as PMMA) improve the tensile strength and Young’s modulus of the blend [9].

Due to its innate brittleness, HA is not used for large implant applications but only as small unloaded implants or surface coatings on metal implants. HA coatings on the implant surface can (i) achieve bone mineralization [10], (ii) improve the growth of bone, and (iii) allow for greater direct contact of the implant surface with the tissue, while the absence of HA cannot create sufficient direct bone deposition or apposition on the titanium surface [11].

Methods such as thermal or plasma spraying [12,13,14], exposure to simulated body fluid [15,16,17], sol–gel [18,19], and electrophoretic [20,21,22,23] and electrochemical cathodic deposition [24,25,26] are used to coat HA. Of these techniques, the cathodic method, is an easy and inexpensive way of obtaining the calcium phosphate (or HA) coating directly on the surface without any post-treatment at room temperature. By maintaining a calcium to phosphorus molar ratio of 1.67 in the electrolyte, HA can be directly obtained on the titanium cathode at the specific current density.

Bone is composed of nano-crystalline HA crystals in a collagen matrix. Synthetic nano-structured HA can increase osteoblast functions [27] and biocompatibility for the microvascular endothelium [28]. Nano-crystalline HA can be obtained on titanium alloy surfaces as a coating, and its bone-bonding characteristics may be improved [29,30]. It is also possible that a calcium phosphate coating can be obtained on a titanium substrate by cathodic deposition and post-treated (annealing or hydrothermal treatment) to obtain HA.

Cathodic deposition of nano-crystalline HA on a Ti6Al4V substrate was reported earlier using an electrolyte containing dissolved compounds of calcium and phosphorus [31]. Electrolysis was carried out at different current densities using ultrasonic stirring or agitation of the electrolyte and reported in [32]. In the literature there are reports of the electrochemical (cathodic or electrophoresis) deposition of hydroxyapatite on different substrates (titanium or magnesium or cobalt–chromium, or stainless steel) through different variations (normal or pulsing or ultrasonic agitation) [19,20,21,22,23,24,25,33].

Ultrasonics is used for fabricating components [34] or for different surface modification techniques [35,36,37,38,39,40]. However, corrosion aspects of the ultrasonically produced hydroxyapatite coating on titanium have been very scarcely reported. The current article deals with this novel and important aspect of research. Structural features of HA coatings obtained from ultrasonic agitation were reported earlier [32], and the corresponding corrosion behavior is discussed in the current article.

## 2. Materials and Methods

The electrolyte was prepared using 42 mM Ca(NO_3_)_2_ and 25 mM (NH_4_)_2_HPO_4_ (Ca:P molar ratio being 1.67) in aqueous medium. The bath had a starting pH of 4.1. Ti6Al4V samples of area 2 × 3 cm^2^ formed the cathode in all the experiments. A stainless-steel plate (area of 6 × 3 cm^2^) formed the anodic part of the electrolyte cell. Sequential operations, such as (i) surface polishing in a series of metallographic papers from rough to smooth, and (ii) etching the polished surface in a solution made from HF:HNO_3_:H_2_O in a ratio of 1:4:5, were done before the actual deposition. Coatings were obtained by electrolysis for 30 min under each of these 3 conditions: (a) 20 mA/cm^2^ with ultrasonic agitation, (b) 50 mA/cm^2^ with ultrasonic agitation, and (c) ultrasonics at 50 mA/cm^2^ without ultrasonic agitation.

A corrosion test was conducted in simulated body fluid, as proposed by Cigada [41]. It was prepared by adding NaCl 8.74 g, NaHCO_3_ 0.35 g, Na_2_ HPO_4_ 0.06 g, and NaH_2_PO_4_ 0.06 g in 1000 mL of H_2_O with an initial pH of 7.4. Using a microprocessor-controlled potentiostat (ACM Gill, UK), potentiodynamic polarization was carried out. The calcium phosphate coating served as the working electrode, and a graphite rod was used as the counter electrode, and a saturated calomel electrode (SCE) was the reference electrode. Polarization was conducted by scanning in the potential range of −700 to 4000 mV at 2 mV/s. EIS was tested in the frequency range of 10^4^ to 10^−2^ Hz, and the results were recorded. An equivalent circuit model was also proposed for this behavior.

## 3. Results and Discussion

### 3.1. Structure

The developed coatings were identified with names for easier reference, and these details are provided in Table 1. Details of the structures of the coatings were discussed in our earlier work [32] using XRD, SEM, and TEM.

Briefly, the coating 1us had an acicular morphology of HA typical of ultrasonic agitation, with the grains being in the size range of 50–100 nm. The coating had a thickness of 100 µm and contained 87% dicalcium phosphate dihyrate (DCPD; Ca:P ratio of 1:1) and 13% HA (Ca:P ratio of 1.67:1). The coating 2hcd was produced at 50 mA/cm^2^ without the use of ultrasonic agitation. This coating contained tricalcium phosphate (Ca:P ratio of 1.5:1) and not HA. However, the coating was thick (200 µm), dense, and amorphous as well [32]. Coating 3ushcd contained 60% DCPD and 40% HA [32]. The coating was thick (300 µm) and had dense deposits. This could be because of the enhancement in deposition arising out of the ultrasonated bath as well as the higher cathodic current density. Electron diffraction rings indicated HA with grains in the range of 50–100 nm [32]. Coatings produced from the electrolyte used in this article yielded HA directly, although current density and ultrasonic agitation played a significant role in this formation. Reactions occurring at the cathode were (i) hydrogen evolution, (ii) oxygen reduction, (iii) hypophosphite reduction, and (iv) HA formation. Based upon the electrolyte pH and cathodic current density, applied HA or other types of calcium phosphates were formed when the phosphate ions reacted with calcium and hydroxide ions [42].

Ultrasonic agitation of the electrolyte deposited calcium phosphate in the grain size of nanometers. At the same time, this also enhanced the rate of competing hydrogen gas evolution. While it evolved, this hydrogen removed the calcium cations away from the cathode. Therefore calcium-depleted coatings (Ca/P ratio close to 1) were produced on the cathode Ti6Al4V.

At the current density of 50 mA/cm^2^, more calcium ions were brought to (or near) the cathode. Very dynamic equilibrium was established between (i) hydrogen evolution and (ii) calcium ion availability. The increased availability of calcium levels increased the Ca/P ratio of the coatings, and HA (Ca/P ratio close to 1.67) was formed [32], and the crystallinity and thickness of the coating was increased.

### 3.2. Corrosion

Each coating was tested for corrosion (a) in the as-plated condition and (b) after every day, for up to four days of continuous exposure to SBF. Figure 1, Figure 2 and Figure 3 show the potentiodynamic polarization curves for the coatings ‘1us’, ‘2hcd’, and ‘3ushcd’ for different conditions.

The polarization curves were obtained for the as-coated condition and exposure to SBF for 12 h, 1 day, 2 days, and 3 days. The polarization curves of the three samples moved towards higher currents upon exposure to SBF. Table 2 shows the results of polarization.

The coating 3ushcd did not show any tendency to passivate at all the test times. However, it showed ennoblement of corrosion potentials (Figure 4) and lower corrosion currents (Figure 5) with increasing time of exposure.

This property could be attributed to the presence of the increased quantity of calcium phosphates on the coating 3ushcd. Phosphates can form gel-like protective layer on anodic titanium oxides and prevent the attack of oxide coating or bare metal beneath by the environment [43]. This deposit covers the entire surface and blocks the cathodic reaction. Since the cathodic current is reduced, the anodic current is also be reduced, causing an overall reduction in the corrosion rate and an increase of the polarization resistance at the ocp [44].

Figure 6, Figure 7 and Figure 8 give the SEM images of the surfaces of 1us, 2hcd, and 3uscd, respectively, after the electrochemical polarization test. The coatings showed cracks and delamination. It is possible that the simulated body fluid (SBF) would have filled up the cracks and formed a gel-like layer with increased time of exposure. This gel resists further attack manifested by ennoblement of corrosion potentials or a decrease of corrosion current values.

Table 3 gives the results of EIS tests. The coating 1us showed inferior impedance at all the times compared to the other two coatings. Figure 9 provides the equivalent circuit modeling of the corrosion of coating 1us. A basic R–C element was connected parallel to a constant phase element (CPE) and in series with Warburg impedance.

CPE is concerned with the electron transfer events occurring at the interface and is influenced predominantly by the porous nature of the coating. Thus, CPE may represent the impedance arising due to surface roughness or inhomogeneities. Warburg impedance represents a Faradaic process, which refers to ion transfer events taking place across the coating. It could be inferred from Table 3 that the total impedance offered by the coatings was actually the impedance contributed by CPE. This shows the importance of surface (coating) roughness or inhomogeneities in establishing the nature of corrosion.

The EIS data had to be fitted with equivalent electronic circuit models. Figure 10 shows one model to explain the corrosion of 2hcd and 3ushcd. One R–C element was connected in series with CPE as well as Warburg. Impedance values were higher in these coatings compared with those of 1us. Accordingly, the equivalent circuit of these two coatings contained Warburg and CPE connected in series with the R–C element.

Crystallization produces paths for electron conduction and allows for anodic reactions. These would permit dissolution of the coating. Of the three coatings developed, the 2hcd coating had amorphous regions and should have slowed the anodic reactions and should therefore have shown high resistance to corrosion. However, it showed moderate rates of corrosion. It could be construed that crystallization effects did not have a significant influence on the coating’s dissolution or corrosion behavior.

Corrosion resistances of the coatings produced in this study had higher E_corr_, lower i_corr_, and higher R_ct_ compared to the cathodic hydroxyapatite coatings produced on magnesium alloy by ultrasonic agitation of the electrolyte [38]. Especially the impedance of 2hcd and 3ushcd coating were three orders higher than the impedance recorded by the HA on the Mg alloy [38]. This shows the extent of corrosion resistance produced by the current coatings reported in this article.

The corrosion potential of 3ushcd was comparable to that of grapheme oxide-HA coating made on treated titanium using ultrasonic agitation [36], while the corrosion current was higher in the present case. This is because of the protective nanotubular layer formed in the pre-treatment [36], which offers resistance to corrosion.

With respect to the crystalline 3ushcd, there are more electron conductive paths and an increased rate of anodic reactions. Therefore, this coating should have exhibited higher corrosion current, but it behaved otherwise. The 3ushcd coating is a thick coating with different kinds of calcium phosphates. A greater thickness could have provided a difficult path for the diffusion of corroding species across the coating and hence may have limited the electrochemical reactions. Among the calcium phosphates, DCPD is highly resorbing compared to HA. Hence, it showed a higher dissolution tendency than did HA. Coating 1us had 87% DCPD, while the coating 3ushcd contained 60% DCPD, This explains the inferior corrosion resistance of 1us, as it had a greater amount of DCPD. The coating 3ushcd had lower DCPD content and hence showed lower resorption ability. For these reasons, 3ushcd showed overall lower rates of corrosion.

## 4. Conclusions

Calcium phosphate coatings are produced on Ti6Al4V by cathodic deposition with the electrolyte being agitated by ultrasonic vibration. The process conditions involved are 20 mA/cm^2^ and 50 mA/cm^2^ with or without ultrasonic agitation of the electrolyte. Ultrasonic agitation and higher current density contribute to the deposition of the HA phase among other calcium phosphates. Polarization and EIS are used to estimate corrosion of the coatings in simulated body fluid (SBF). The coating produced at 50 mA/cm^2^ from an ultrasonated bath has very good resistance to the corrosive attack by SBF. Higher crystallinity and coating thickness contribute to the increased corrosion resistance of this coating.

## Figures and Tables

**Figure 1 materials-15-08609-f001:**
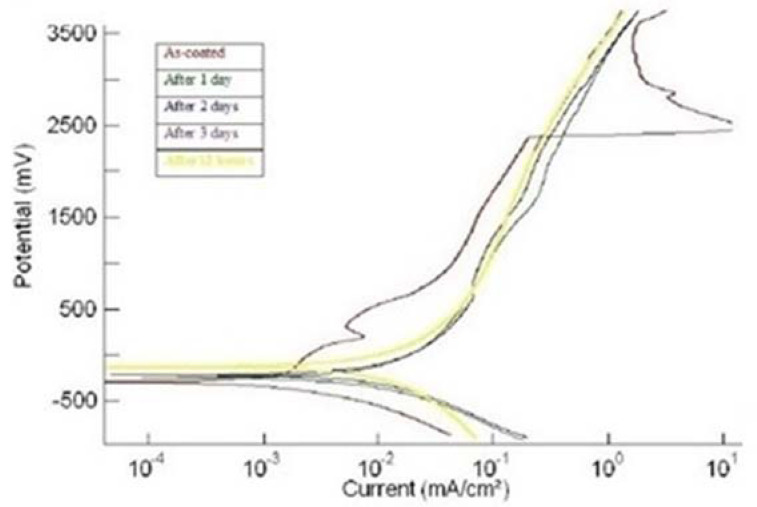
Polarization curves of ‘1us’.

**Figure 2 materials-15-08609-f002:**
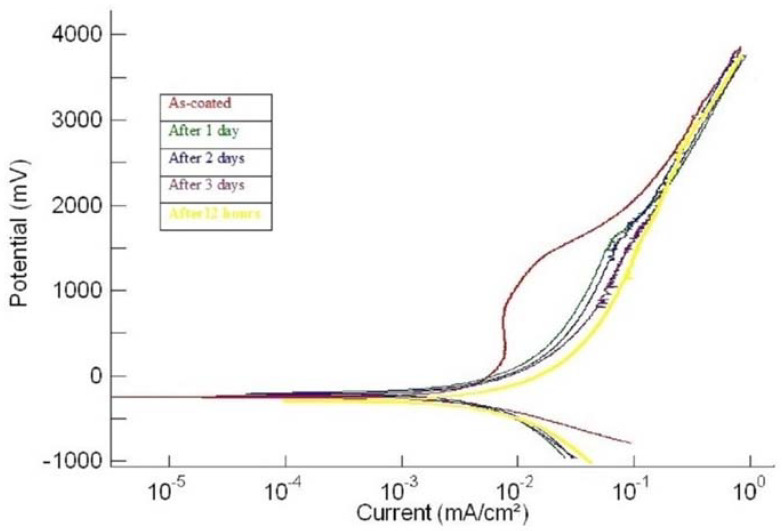
Polarization curves of ‘2hcd’.

**Figure 3 materials-15-08609-f003:**
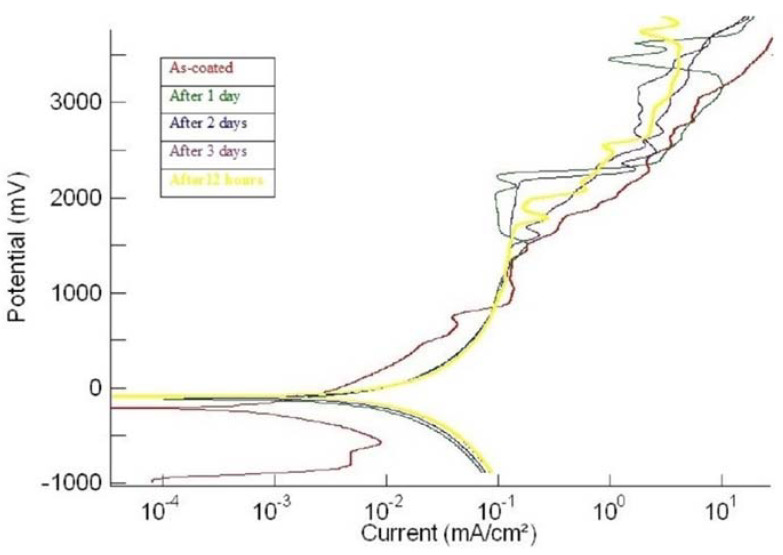
Polarization curves of ‘3ushcd’.

**Figure 4 materials-15-08609-f004:**
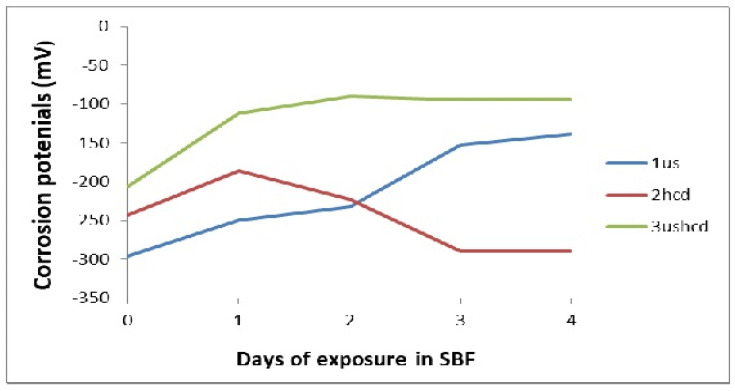
Variation of corrosion potentials with time of exposure.

**Figure 5 materials-15-08609-f005:**
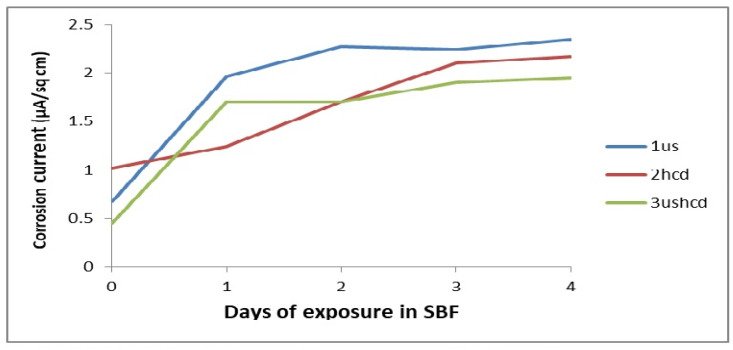
Variation of corrosion currents with time of exposure.

**Figure 6 materials-15-08609-f006:**
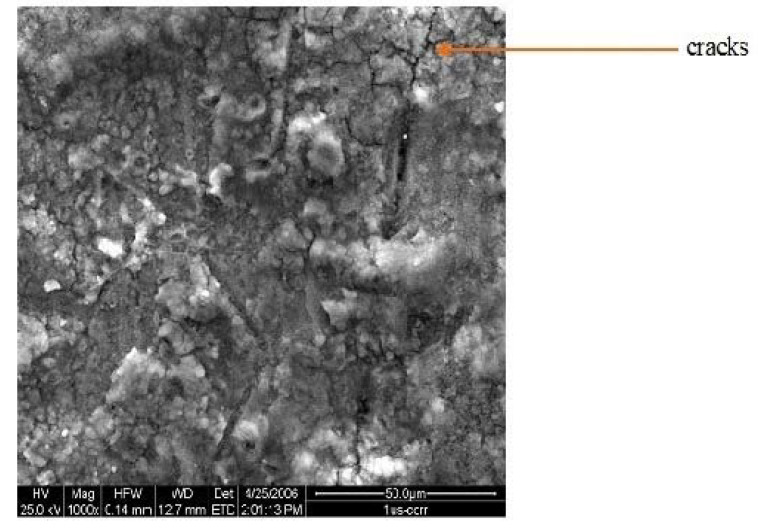
SEM of surface of 1us after polarization test in SBF.

**Figure 7 materials-15-08609-f007:**
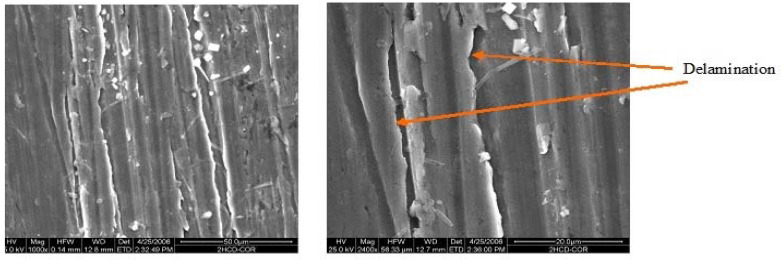
SEM of surface of 2hcd after polarization test in SBF.

**Figure 8 materials-15-08609-f008:**
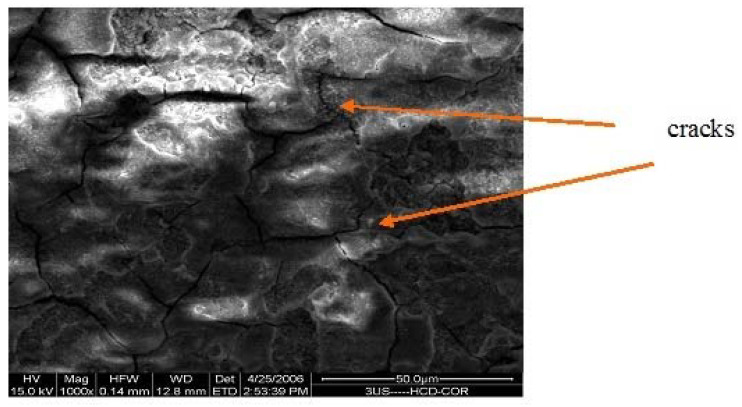
SEM of surface of 3ushcd after polarization test in SBF.

**Figure 9 materials-15-08609-f009:**
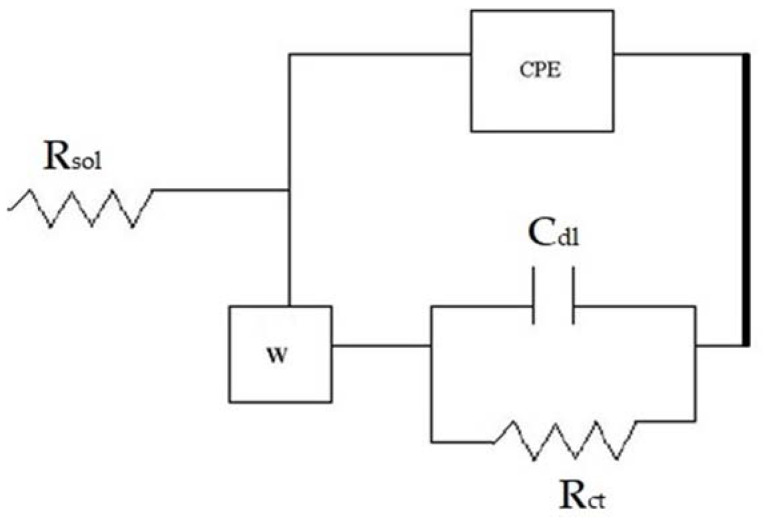
Equivalent circuit depicting the corrosion of 1us.

**Figure 10 materials-15-08609-f010:**
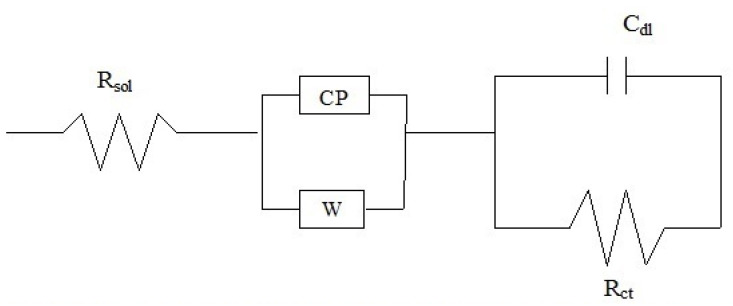
Equivalent circuit depicting the corrosion of coatings 2hcd and 3ushcd.

**Table 1 materials-15-08609-t001:** Nomenclature of the coatings.

Electrolyte	Electrolysis Conditions	Further Treatment	Nomenclature
0.042 M Ca(NO_3_)_2_ and 0.025 M (NH_4_)_2_HPO_4_	20 mA/cm^2^ 30 min	Ultrasonic agitation	1us
0.042 M Ca(NO_3_)_2_ and 0.025 M (NH_4_)_2_HPO_4_	50 mA/cm^2^ 30 min	-	2hcd
0.042 M Ca(NO_3_)_2_ and 0.025 M (NH_4_)_2_HPO_4_	50 mA/cm^2^ 30 min	Ultrasonic agitation	3ushcd

**Table 2 materials-15-08609-t002:** Results of electrochemical polarization.

Coating	Testing Condition	E_corr_ (mV)	i_corr_ (µA/cm^2^)	E_pass_ (mV)	i_pass_ (µA/cm^2^)
1us	(i) As-plated	−296	0.67	190–350	7.5
	(ii) After 1 day	−250	1.96	No passivation	No passivation
	(iii) After 2 days	−233	2.27	No passivation	No passivation
	(iv) After 3 days	−153	2.24	No passivation	No passivation
	(v) After 4 days	−139	2.35	No passivation	No passivation
2hcd	(i) As-plated	−243	1.02	250–950	7.7
	(ii) After 1 day	−186	1.24	No passivation	No passivation
	(iii) After 2 days	−223	1.7	No passivation	No passivation
	(iv) After 3 days	−290	2.10	No passivation	No passivation
	(v) After 4 days	−290	2.17	No passivation	No passivation
3ushcd	(i) As-plated	−207	0.45	No passivation	No passivation
	(ii) After 1 day	−112	1.7	No passivation	No passivation
	(iii) After 2 days	−95.0	1.7	No passivation	No passivation
	(iv) After 3 days	−90.0	1.9	No passivation	No passivation
	(v) After 4 days	−95.0	1.95	No passivation	No passivation

**Table 3 materials-15-08609-t003:** Results of ac impedance.

Coating	Warburg Imp. (Ώ-cm^2^)	R_ct_ (Ώ-cm^2^)	C_dl_ (F)	Z_CPE_ or Total Imp. ‘Z’ (Ώ-cm^2^)
1us				
(i) As-plated	4.9 × 10^8^	1.44 × 10^5^	2.61 × 10^−4^	8.19 × 10^5^
(ii) After 1 day	2.6 × 10^8^	0.39 × 10^5^	2.61 × 10^−4^	5.79 × 10^5^
(iii) After 2 days	1.5 × 10^8^	0.76 × 10^5^	2.61 × 10^−4^	5.35 × 10^5^
(iv) After 3 days	1.42 × 10^8^	1.44 × 10^5^	2.61 × 10^−4^	15.5 × 10^5^
(v) After 4 days	1.45 × 10^8^	0.54 × 10^5^	2.61 × 10^−4^	2.5 × 10^5^
2hcd				
(i) As-plated	4.0 × 10^5^	5.42 × 10^5^	2.61 × 10^−4^	1.69 × 10^8^
(ii) After 1 day	2.86 × 10^5^	1.61 × 10^5^	2.61 × 10^−4^	20.7 × 10^8^
(iii) After 2 days	2.65 × 10^5^	1.67 × 10^5^	2.61 × 10^−4^	2.087 × 10^8^
(iv) After 3 days	2.59 × 10^5^	1.94 × 10^5^	2.61 × 10^−4^	2.53 × 10^8^
(v) After 4 days	2.55 × 10^5^	1.31 × 10^5^	2.61 × 10^−4^	1.49 × 10^8^
3ushcd				
(i) As-plated	89.6 × 10^5^	2.85 × 10^5^	2.61 × 10^−4^	5.33 × 10^8^
(ii) After 1 day	2.82 × 10^5^	0.76 × 10^5^	2.61 × 10^−4^	8.63 × 10^8^
(iii) After 2 days	29.1 × 10^5^	1.17 × 10^5^	2.61 × 10^−4^	1.35 × 10^8^
(iv) After 3 days	2.22 × 10^5^	1.17 × 10^5^	2.61 × 10^−4^	17.2 × 10^8^
(v) After 4 days	2.31 × 10^5^	1.37 × 10^5^	2.61 × 10^−4^	63 × 10^8^

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
