# Peer review of "Corrosion of Stirred Electrochemical Nano-Crystalline Hydroxyapatite (HA) Coatings on Ti6Al4V"

_materials, 2022, doi:10.3390/ma15238609_

Round 1
Reviewer 1 Report
I had reviewed the manuscript entitled " Corrosion of stirred electrochemical nano-crystalline hydroxy-apatite (HA) coatings on Ti6Al4V. The current work, in my opinion, is very interesting and novel. In my opinion, the following points should be considered for further improvement:
(1) In the introduction, authors should clearly state the novelty of their research.
(2) Please cite the following paper regarding bone growth for the benefits of readers
Timimi, Z., & J Tammemi , Z. . (2022). Polymer Blends and Nanocomposite Materials Based on Polymethyl Methacrylate (PMMA) for Bone Regeneration and Repair: Characterization and Preparation. Journal of Sustainable Materials Processing and Management, 2(1), 15–23. Retrieved from https://publisher.uthm.edu.my/ojs/index.php/jsmpm/article/view/10674
(3) Authors should also explain to the reader, the benefits of nano coating.
(4) In section 3.1, Authors should provide a brief description of their previous discovery [27]. Authors should provide readers with a summary of the XRD, SEM, and TEM results, such as crystallinity, size, thickness, phase, and Ca/P ratio of HA nano-coatings.
(5) In section 3.2, authors should compare their results with previous studies and make critical comments on their comparasion
(6) The quality of images should be improved.
Reviewer 2 Report
This paper deals with a much studied subject (on scopus there are 157 answers at "hydroxyapatite AND corrosion AND ultrasonic" but which needs a lot of more research. Indeed this makes it well within the theme of Materials.
However it is not acceptable in its current form.
Concerning the EIS results, it is necessary to give the experimental Nyquist or/and Bode diagrams and to precise the standard error of the fitted data which are provided. Without these values, there is no confidence in the data. More, the units of the impedances have to be corrected.
Some assumptions about the surface deposits formed during corrosive bath should be confirmed by further physico-chemical analysis such as EDX or XPS
Reviewer 3 Report
Refereree report on manuscript “Corrosion of stirred electrochemical nano-crystalline hydroxyapatite (HA) coatings on Ti6Al4V”
This version does not look worthy and cannot be recommended for publication in this form and at least needs major revision.
1. Introduction. Almost all of the first 20 references are older than 10 years. Because of this, the motivation and relevance of this study may be in doubt, because the latest achievements in HA research and development are not reflected. This information should be updated, for example, using the search for the latest achievements: https://www.mdpi.com/search?q=hydroxyapatite
2. Introduction. Line 41. It is necessary to note their resistance to both ultraviolet and X- irradiations. The outcome of HA indeed depends on their resistance to aging, including radiation. See:
Bystrova, A.; Dekhtyar, Y.D.; Popov, A.; Coutinho, J.; Bystrov, V. Modified hydroxyapatite structure and properties: Modeling and synchrotron data analysis of modified hydroxyapatite structure. Ferroelectrics 2015, 475, 135–147.
Hübner, W.; Blume, A.; Pushnjakova, R.; Dekhtyar, Y.; Hein, H.-J. The influence of X-ray radiation on the mineral/organic matrix interaction of bone tissue: An FT-IR microscopic investigation. Int. J. Artif. Organs 2005, 28, 66–73.
3. The declared nanocrystallinity requires a clear and detailed explanation
4. Line 88. Why is declared amorphous state not checked directly?
5. Lines 86 -95. It is difficult to follow here, where are the literature data, and where are the experimental results? To what extent old results can be used in this work?
6. Why are legends for figures, both vertical and horizontal axes, made using different fonts and sizes?
7. In the conclusions, it is necessary to clearly formulate what new data about the studied materials were obtained in this work? This is somehow connected with the relevance and novelty of research, which, unfortunately, is not disclosed at all in the introduction.
In general, the manuscript is interesting and can be considered for publication after constructive reflection on the above comments.
Round 2
Reviewer 3 Report
The authors have successfully performed the revision of their original manuscript, so now it can be recommended for publication